# Effects of Physical Exercises and Verbal Stimulation on the Functional Efficiency and Use of Free Time in an Older Population under Institutional Care: A Randomized Controlled Trial

**DOI:** 10.3390/jcm9020477

**Published:** 2020-02-09

**Authors:** Agnieszka Wiśniowska-Szurlej, Agnieszka Ćwirlej-Sozańska, Natalia Wołoszyn, Bernard Sozański, Anna Wilmowska-Pietruszyńska

**Affiliations:** 1Institute of Health Sciences, Medical College of Rzeszow University, 35-310 Rzeszow, Poland; sozanska@ur.edu.pl (A.Ć.-S.); natalia.woloszyn@op.pl (N.W.); benieks@poczta.onet.pl (B.S.); 2Faculty of Medicine, Lazarski University, 02-662 Warsaw, Poland; anna.wilmowska@autograf.pl

**Keywords:** sedentary behaviour, aged, exercise, motivation

## Abstract

Older people in institutional care are, for the most part, physically inactive and do not interact with each other or medical staff. Therefore, reducing sedentary behaviour is a new, important, and modifiable lifestyle variable that can improve the health of elderly people. The aim of the project was to assess the degree of improvement in functional performance and the possibility of changing habitual, free time behaviour among elderly people under institutional care by applying physical training with verbal stimulation. The study covered older people, aged 65–85 years, who are living a sedentary lifestyle in care homes in Southeastern Poland. Those who met the eligibility criteria were enrolled in the study and were assigned, at random, to one of four parallel groups: basic exercises (*n* = 51), basic exercises combined with verbal stimulation (*n* = 51), functional exercise training (*n* = 51), and functional exercise training with verbal stimulation (*n* = 51). No statistically significant differences in baseline characteristics were observed across the groups. Data were collected at baseline and at 12 and 24-weeks following the completion of the intervention. In the group with functional exercise training with verbal stimulation, in comparison to the group with basic exercises, the greatest positive short-term impact of intervention was demonstrated in terms of functional fitness (increased by 1.31 points; 95% confidence interval (CI) = 0.93–1.70), gait speed (improved by 0.17 m/s, 95% CI = 0.13–0.22), hand grip strength (by over 4 kg; 95% CI = 2.51–4.95), and upper-limb flexibility (by 10 cm; 95% CI = 5.82–12.65). There was also a significant increase in the level of free-time physical activity and an improvement in the quality of life, especially as expressed in the domain of overall physical functioning. Our study showed that a functional exercise program, combined with verbal stimulation, is effective at improving physical fitness and raising the level of free-time physical activity.

## 1. Introduction

In recent years, there has been a slowdown in the demographic development of people in Europe and Poland. At the end of 2017, the number of people aged 65 and older in Poland amounted to six million, of which over 105,000 were covered by institutional care [1,2]. While changes in care and health policies and services have propagated alternative, long-term care methods, around 25% of older people spent the last years of their lives in a care home [3].

A low level of physical activity (PA) is one of the most serious health problems facing older people [4]. A lack of PA increases the incidence of cardiovascular diseases, reduces cardiovascular and respiratory function, and increases the risk of falling, osteoporotic fractures and disability [5]. According to a report published by the American Journal of Clinical Nutrition, a sedentary lifestyle accounts for twice as many deaths in Europe as obesity. Moreover, increasing PA levels among Europeans would reduce the deaths in Europe by 7.5%, as reported by Ekelund et al. [6]. Encouraging even a slight increase in activity, with reference to inactive people, can be beneficial to public health.

An insufficient level of PA is one of the main behavioural burdens in the world [7]. From 60% to 70% of older people do not meet the World Health Organization’s recommendations, with respect to PA, to achieve health benefits [8]. The nationwide Polish data compiled by Kozdroń show that only 7% of people aged 60–64 and 0.6% aged 80 years and more undertake regular physical activity [9]. Research conducted in nursing homes indicates that their residents are physically inactive most of the time and do not interact with each other or medical personnel [10].

There are a number of factors in the daily life of older people that can hinder PA. These include, among other things, a negative exercise history, lack of exercise skills, low social and cultural support, and a low level of motivation and self-efficacy [11]. According to Resnick et al. [12], interventions that focus on teaching older people about the benefits of PA, setting goals for exercise program implementation, and reducing unpleasant exercise-related sensations can improve regular participation in exercises and functional performance [13]. Another important element influencing whether an older person is able and willing to participate in physical activity and a model of healthy behaviours is motivation. Motivation is a term that defines the process regulating behaviour that is designed to control the engagement, maintenance, and termination of activities.

Despite the publication of several systematic reviews concerning the effectiveness of exercise programs for older people, the most effective of them has not been clearly identified yet. A systematic review of Carode et al. [14] showed that a multi-component intervention is the best strategy to improve the functional state of older people. Marcos-Pardo et al. developed a motivational resistance-training program and demonstrated the positive effect of training on motivational variables and the body composition of older people. However, the described results concern a preliminary study with a small number of people affected by the intervention [15]. In a systematic review, Farrance et al. indicated that programs for group exercises that used social support proved to be an effective way to improve physical fitness and to increase the level of PA for older people [16]. The authors noted that incorporating education about exercise benefits, social ties, and professional advice on enhancing health behaviours, with reference to exercise programs, could provide guidelines for designing innovative interventions in order to improve the health of older people. The International Association of Gerontology and Geriatrics (IAGG) Global Aging Research Network and the IAGG European Region Clinical Section have developed recommendations regarding motivation and pleasure as key factors to increase the level of PA in people receiving long-term care [17]. Experts have argued for the development and implementation of long-term care improvement strategies to improve the health of residents. Therefore, the aim of the project was to assess the degree of improvement in functional performance and the possibility of changing habitual ways of spending free time among elderly people under institutional care, by applying physical training with verbal stimulation. We hypothesized that the biggest changes regarding the functional efficiency and use of free time will be observed in the group with functional exercise training connected to verbal stimulation.

## 2. Materials and Methods

### 2.1. Trial Design

Our study contained a controlled, randomized trial with four open-label parallel groups. The study was conducted from March 2016 to December 2018. Data were collected at baseline and at 12 and 24-weeks. This procedure was established according to the “CONSORT” statement [18]. The study was registered in the Sri Lanka Clinical Trials Registry (SLCTR / 2016/004).

Availability of data and materials: All data used in this study were stored at https://repozytorium.ur.edu.pl/handle/item/5094.

Ethics approval and consent to participate: The research project was accepted by the Bioethics Committee of the University of Rzeszow (Resolution No. 6/06/2015). In accordance with the Declaration of Helsinki, the participants were provided with information about the aim and the course of the study, and expressed their written and informed consent to participate. The older persons were informed about the possibility of withdrawing from the study at any point during the study procedures.

### 2.2. Participants

The study was conducted in nine randomly selected nursing homes for older people and the chronically physically ill in Southeastern Poland. The management department of the centers were informed about the objectives and conduct of the study. After obtaining permission to implement the project in individual centers, the recruitment of participants began. To promote the exercise programs, information was posted, leaflets were distributed, and residents were orally informed about the details. The initial qualification for participants in the study was made by a physiotherapist working in the center. Participants who were eligible for the trial were required to comply with the following criteria: age 65–85 years, Mini-Mental State Examination (MMSE) score >19, Geriatric Depression Scale (GDS) score <11 points, physical fitness without serious restrictions, Short Physical Performance Battery score >5, and spending free time sitting for at least 4 h/day, 6 to 7 days/week (the Physical Activity Scale for the Elderly questionnaire). Exclusion criteria were: symptoms of cardiovascular diseases, severe systemic disease, severe circulatory or organic insufficiency, severe neurological disorder, injuries of the lower limbs during the last 6 months, use of medication significantly affecting the body’s balance and participation in improvement exercise programs in the 3 months prior to the trial. Subsequently, an interview was conducted with a physician employed in the nursing home to eliminate the health contraindications in performing physical exercise by selected residents. After taking into account the inclusion criteria and obtaining written consent from the physician and residents to participate in the study, 204 people were included in the exercise program

### 2.3. Interventions

The subjects were assigned, at random, to one of four groups:

Group BE: Basic exercises without verbal stimulation

Group BE + VS: Basic exercises combined with verbal stimulation

Group FET: Functional exercise training without verbal stimulation

Group FET + VS: Functional exercise training with verbal stimulation

#### Exercise Program

All study participants took part in a 12-week exercise program, with or without verbal stimulation. Exercises were conducted in groups of 4–8 people, twice a week for 30 min. Each session was adapted to the functional capabilities of the subjects and was supplemented with breathing exercises. The exercise intensity was moderate, at 11–13 points, according to the Borg scale. The load during resistance training was up to 60% of one repetition maximum. Participants exercised with the music preferred by the subgroup (low volume level of 60 dB, moderate tempo). Exercise programs were performed by 9 physiotherapists (one physiotherapist per care home) with at least 2 years of experience working with older people. Each therapist was trained in the field of exercise programs and verbal stimulation before starting the program. They also dealt with the assessment of participants in the exercise programs, and monitored possible pain or other complications resulting from the performed exercises. Their observations were noted in activity diaries prepared for each group. In the case of health problems, such as malaise—a pain caused by excessive effort—the physician supervising the research made the decision to exclude participants from the study. Furthermore, the blood pressure and pulse of the older people were monitored before and after exercise. In addition, the study director was in contact with all physiotherapists in the course of the intervention in order to ensure that the exercise was of high quality and consistent in all care homes.

Basic exercises: the program included exercises performed in a sitting position. The exercises contained elements of aerobic fitness, stretching and equivalent exercises.

Functional exercise training: the program was divided into two sessions. Session I contained strengthening and stretching, with exercises performed in a sitting position using a Thera-Band and gymnastic sticks. Lower-limb strengthening exercises included tasks such as dorsi flexion and plantar flexion of the ankle joints, flexion and extension of the knee joints, flexion, extension and abduction in the hip joints, standing up and sitting down on a chair, and lifting objects from the floor. Upper-body exercises included stretching and strengthening of the shoulder girdle. Session II contained equivalent and functional training, using exercises performed in a seated or standing position, using chairs as stabilizing aids. Functional exercises included complex motor tasks, such as head rotation when sitting and standing, changing the body position, and lifting objects from the floor and keeping them. Balance exercises included exercises performed with and without visual control; static and dynamic balance of the elderly people was practiced (physical support was provided by a physiotherapist and auxiliary aids). The progression of the balance exercises was achieved by reducing the support plane, or by eliminating the visual control. Exercises were oriented towards achieving the goals of functional activity established by the elderly people.

Verbal stimulation: a model physical exercise program that incorporates the use of verbal stimulation. The model is focused on studying the achievements of participant’s in regard to (objective) functional goals, and altering his/her (subjective) perception of the intensity of the training program and their individual aims. Before starting the intervention, a systematic review was performed to assess which elements had the greatest impact on health benefits among older people. The next step was to determine the baseline assessment; therefore, both the short- and long-term objectives of exercise participation were established. Short-term objectives assessed what a person is able to perform while exercising daily (e.g., perform 20 sit-ups, which improves lower extremity strength), whereas long-term objectives determined the “ultimate goal” (e.g., walking to the shop unaided or generally obtaining an improved level of mobility). Furthermore, specified objectives were helpful in achieving individual goals, as well as in persevering with these new health behaviours. According to the results of the studies carried out by Park et al., setting functional goals is an important motivating factor for PA performed by older people [19].

The program uses:Generating extrinsic motivation: the participants were informed about a small reward at the end of the program. Seniors were involved in a given activity to achieve extrinsic consequences. After performing physical exercises, a meeting was organized, at which diplomas for participation in the exercises and commemorative photos were handed out.Generating intrinsic motivation: a model for class management was developed, which assumed that the participants would achieve the set goals (challenges). The subjective feelings associated with performing physical exercises and independent steps towards a goal were strengthened. In accordance with the assumptions of the theory of competence-based models of intrinsic motivation [20], the examined person best estimates, depending on individual beliefs based on the significance of the goals established by them, the possibilities of motivation

In addition, informative materials were provided to the subjects:Explanation of the importance of physical activity in order to maintain independence and health;illustrations of exercises that older people could perform on their own after completing the program;description of fears and barriers appearing while physical activity is undertaken by older people and a description of the necessary ways to overcome them.

According to Locke, education and discussion about the anxiety connected to physical activity allows an older person to feel the need to participate in exercises [21].

The verbal stimulation program was included in the individual exercise programs.

### 2.4. Outcome Measures

Outcome assessments were performed at the baseline and at 12 and 24-weeks. The study was divided into two stages. On the first day, sociodemographic data and questionnaire interviews were collected; on the second day, anthropometric measurements and functional activity tests were performed.

Data regarding age, sex, education, marital status, and years in a nursing home were gathered on the basis of the records kept by care homes and by interviews with the individuals researched. Data related to chronic diseases and the number of drugs were collected from medical records kept by physicians in care homes. Cognitive status was also assessed by MMSE [22] and depression symptoms were assessed by the use of GDS [23].

#### 2.4.1. Main Outcome

The Short Physical Performance Battery (SPPB) test was used to assess the functional status of the participants. The test comprised the assessment of three physical sequences: maintaining balance in three positions, gait speed over a short distance and attempting to stand up from a chair five times without the use of the upper limbs [24].

#### 2.4.2. Secondary Outcomes

##### Physical Activity Assessment

Physical activity was assessed by means of the Physical Activity Scale for the Elderly (PASE) [25]. It is based on the evaluation of how free time was spent, as well as the work-related activities or voluntary work performed within 7 days of the survey, taking into account the frequency, duration and the level of intensity of these activities.

##### Functional Assessment

The performance of basic and complex everyday activities was assessed using the activities of daily living and instrumental activities of daily living (ADL-IADL) scale [26].

##### Muscular Strength Assessments

Hand grip strength (HGS) was carried out by the use of a hand dynamometer (JAMAR PLUS + Digital Hand Dynamometer, Patterson Medical). The values were obtained from measurements carried out on the participants while seated on a chair without armrests, with their feet rested flat on the floor—In pursuance to the recommendations of the American Society of Hand Therapists [27]. The average of the three measurements was recorded. The 5× Sit to Stand Test (5× STS) was used to assess the strength of the lower limbs. The participants were required to stand up 5 times from a sitting position as fast as possible without using their upper extremities [28].

##### Mobility Assessment

Mobility was assessed using the Timed Up and Go test (TUG) without a cognitive task and with a cognitive task (TUG cog) [29]. The TUG test was performed as follows: the participant will stand (from a sitting position on a chair), walk a distance of 3 m, turn around (180°), walk the 3 m back to the starting position, and resume the sitting position. The final result was the average time of the three attempts. Gait speed was assessed using the 10-Meter Walk Test [30]. Participants were asked to walk a distance, marked with an adhesive tape, of 10 m.

##### Flexibility Assessment

The elasticity of the upper and lower limbs was assessed using the back stretch test (BS) and the chair sit and reach test (CSR) [31]. BS estimates the elasticity of the girdle of the upper limb, which is necessary in the course of performing such activities as rubbing and washing the back. The test consisted of griping the hand of one limb from above with the other hand from below and behind the subject’s back. The distance between the fingertips of the middle fingers was measured. CSR assessed the elasticity of the lower body, which was necessary in order to maintain the correct pattern of walking, getting out of the bathtub or dressing socks. The test was to bend the body in a seated position towards the outstretched lower limb. The distance between fingertips and toes was measured.

##### Body Balance Assessment

Balance was assessed using the Berg balance scale (BBS) [32]. The scale included 14 simple tasks, including: changing body position, maintaining a sitting position, maintaining a standing position under visual control and without it, watching, standing in a tandem position, standing on one leg, rotation around the axis, reaching forward, lifting objects from the floor, transferring, and going up a step.

### 2.5. Other Outcomes

#### 2.5.1. Postural Stability Assessment

The assessment of the postural stability was performed by the use of a two-plate stability platform CQ Stab 2P (CQ Elektronik System, Czernica, Poland). Each of the platform plates had 3 force sensors that determined the displacement of the center of pressure on the support plane. During the measurements, the values describing the static balance were recorded. Platform plates were placed parallel, 2 m from the wall of the room where there was a marker to fix eyesight during the test with open eyes. Before each set of measurements was taken, the device was calibrated. The test included a 30 s sample performed with eyes open and eyes closed. The participants were instructed to remove their shoes and take a free-standing position on the platform plates with their arms set adjacent to the torso [33].

All testing procedures were fully explained and presented prior to assessment.

#### 2.5.2. Quality of Life Assessment

Quality of life (QoL) was examined using the SF-36v2 questionnaire, consisting of 36 questions, which analysed the functional profile of health, well-being and psychometric assessment based on the subject’s mental state of health. The quality of life established with respect to physical health was measured using two main domains: functioning in the physical dimension, i.e., general physical health (physical component summary: PCS), and functioning in the mental dimension, i.e., general mental health (mental component summary: MCS) [34].

### 2.6. Sample Size

The sample size was estimated from an a priori power analysis to detect the statistically significant effects of exercise [35]. The sample size was chosen according to the Cohen method, using standard assumptions: 0.05 for significance level, 0.8 for power of test, and 0.5 for effect size, accounting for, according to Cohen, a medium effect size [36]. The sample size calculation for the main outcome measure was based on changes in SPPB scores. The sample size calculation was based on 80% power to detect a one-point change in the SPPB score with an alpha level of 0.05. It was calculated that, based on the final analysis, the total number of people surveyed should amount to 39 people in each group. Therefore, 51 people were recruited to individual groups in order to allow for a 20% dropout rate from the study.

### 2.7. Randomization and Blinding

Randomization that implemented the stratified method by the use of the statistical package R 3.2.2 (The R Foundation for Statistical Computing, Vienna, Austria) was carried out. Four-in-one blocks were randomized, which made it possible to obtain an even distribution of elderly people in the studied groups. The order of randomization was determined by using a computerized schedule of random numbers. An independent biostatistician implemented randomization, hid the block size from the executive module and used randomly mixed block sizes. He was responsible for the confidentiality of the list of people included in the study. Outcome assessors were blind to the group division and did not take part in implementing interventions. Due to the ensemble nature of care homes, participants in the study were not blinded after being assigned to groups.

### 2.8. Statistical Methods

Descriptive characteristics were presented as a mean and standard deviations, or a number and percent when appropriate. A one-way ANOVA test was used to assess the differences between groups. The mean difference between treatment groups and the confidence intervals for quantitative variables was also determined. Post-hoc analysis for the quantitative variables analysis was conducted with *t*-tests with Bonferroni correction. The significance of changes in the examined variables, between two time points, were assessed with paired *t*-tests. Standard intention-to-treat analysis was performed for each outcome. Missing data were deemed to be missing at random and were calculated using the imputation technique according to the protocol study [37]. Analyses were conducted at a 0.05 level of significance. R software version 3.6.1 (The R Foundation for Statistical Computing, Vienna, Austria) was used.

## 3. Results

After the initial test was performed and the inclusion and exclusion criteria were taken into account, the older people were randomly assigned to four exercising groups: BE group, 51 people; BE + VS group, 51 people; FET group, 51 people; and FET + VS group, 51 people (Figure 1). The withdrawal rate from the study was: BE group, 10 people; BE + VS, 12 people; and FET, 10 people and FET + VS, 9 people. The main reasons for withdrawal from the study were: moving house, influenza diagnosed by a physician, refusal to participate without any reason given, and the death of a participant. The analysis included people whose attendance at exercises was over 80%.

In the studied groups, baseline parameters did not differ across the groups in terms of sociodemographic features and clinical parameters, including: cognitive status, mood, functional state, mobility, muscle strength, flexibility, body balance, postural stability with and without visual control, quality of life and level of physical activity. The average age of the study groups fluctuated between 73 and 74 years. Baseline scores of the research variables are presented in Table 1. Postural balance characteristics of the participants can be found in Appendix A.

Regarding the BE group, after 12 weeks of exercises, a statistically significant improvement was noticed in the following areas: functional fitness, muscle strength of lower limbs, mobility and gait speed, and flexibility of the lower limbs. The observed improvement was only maintained for the following 12 weeks in terms of lower-limb flexibility.

In the BE + VS group, after 12 weeks of exercises, a statistically significant improvement was shown in regard to functional fitness, performing complex everyday activities, hand grip and lower-limb strength, flexibility of the upper and lower limbs, body balance, as well as the quality of life of the people studied. The obtained change was maintained in most of the aforementioned parameters until 24 weeks after the exercises began.

The largest positive changes were observed in the groups with FET and FET + VS. In the group FET + VS, the greatest positive short-term impact of intervention was demonstrated in terms of functional fitness (SPPB increased by 1.31 points; 95% CI = 0.93–1.70), gait speed improved by 0.17 m/s (95% CI = 0.13–0.22), hand grip strength enhanced over 3.5 kg (95% CI = 2.51–4.95) and flexibility of the upper limbs developed by 10 cm (95% CI = 5.82–12.65). There was also a significant increase in the level of physical activity spent in free time (an increase by 6.91 PASE score; 95% CI = 4.58–9.24) and an improvement in the quality of life, especially in the domain of overall physical functioning (an increase by 11.79 score; 95% CI = 8.64–14.95).

The groups exercising with a verbal stimulation element were characterized by maintaining improvement in most of the studied parameters for a period of 24 weeks from the beginning of the study (Table 2). A mean difference scores for each group across time in terms of postural balance can be found in Appendix A. 

In order to assess the differences between the four groups (BE, BE + VS, FET, and FET + VS), a one-way ANOVA analysis was applied. After 12 weeks of exercises, there was a statistically significant difference between the BE and FET + VS groups. In the FET + VS group, functional fitness, mobility without a cognitive task, flexibility of the upper and lower limbs, hand grip and lower-limb strength, balance and quality of life in the physical domain significantly improved, in comparison to the BE group (SPPB 0.25 vs. 1.27; TUG −0.91 vs. −3.89; HGS _R_ −0.65 vs. 3.63; 5× STS −2.55 vs. −6.36; BS _R_ 1.57 vs. 9.27; and BBS 1.18 vs. 7.27). All the studied groups after the intervention improved, in a similar way, their ability to perform basic and complex everyday activities, mobility with a cognitive task, and quality of life in the mental domain (Table 3). No statistically significant difference between groups was observed in body balance parameters after 12 weeks of exercise (Appendix A).

After 24 weeks of commencing exercises, the greatest effects were noted in the FET + VS group, in comparison to the BE group in most of the studied parameters besides lower-limb flexibility and quality of life in the mental domain. A significantly stronger short-term impact of interventions in the FET + VS group was demonstrated in the range of improvement of functional fitness and changes in the habit of spending free time among older people (*p* < 0.001). In the FET + VS group, compared to the other exercising groups (BE, BE + VS, FET), there were statistically significant, larger positive changes in the functional fitness, leisure-time activity, performance of complex daily activities, mobility, gait speed, and quality of life in the physical domain (*p* < 0.001) (Table 4). No statistically significant difference between groups was observed in body balance parameters after 24 weeks follow-up (Appendix A).

## 4. Discussion

Observational studies suggest that 97% of daytime is spent sedentarily by residents of nursing homes, e.g., sitting and watching TV with low levels of interaction with each other and with medical staff [38]. A lack of engagement with PA has a detrimental effect on physical and mental health, quality of life and social isolation [39]. Therefore, reducing sedentary behaviour is an important new modifiable variable of lifestyle that can improve the health of older people [40]. According to the Copenhagen Consensus statement (2019) that considers PA and aging, researchers determined that self-efficacy, intentions, and the perceptions of one’s health are related to the person’s level of PA and interventions based on the theory that behavioural changes provides greater results. According to this account, they concluded that future research should assess the potential of these factors to promote PA and the good health of seniors [41].

To the best of our knowledge, this is the first intervention that assesses the impact of exercise programs, combined with verbal stimulation, aimed at motivating people to improve physical fitness and at changing habitual ways of spending free time by older people in institutional care. Furthermore, we have observed that the functional exercise program, combined with verbal stimulation, is the most effective in improving functional fitness, mobility, muscle strength and flexibility, as well as increasing the level of physical activity spent in free time and improving quality of life, especially within the physical domain.

Motivational strategies included in a resistance-training program affected psychological needs, motivation and compliance with the physical-activity principles [15]. Findings on interventions that increased the level of PA in free time are of moderate quality and focus mainly on systematic reviews, indicating a number of guidelines for the techniques used in order to change adult behaviour [42,43]. A systematic review by Orrow et al. [44] showed that the promotion of PA used in older people with a “sedentary” lifestyle leads to a small or medium improvement in the level of physical activity over 12 months. Hilldson et al. demonstrated, in their systematic review, that physical exercise programs moderately affect the functional state of older people and cause changes in the level of PA [45]. They suggested that further research should be planned with a view to propagate the long-term involvement in physical exercises by older people. Taking into account the results of the research herein, and the reports of other authors, when further designing an intervention in order to change the habit of older people in free time and to improve the health and quality of life of older people, it is necessary to focus not only on performing physical exercises at a moderate intensity, but also on the use of verbal stimulation based on generating motivation to replace the time spent sitting down with physical activity.

In our study, the effectiveness of four different exercise programs were compared. Despite the publication of several systematic reviews on the effectiveness of exercise programs for older people, as of yet, the most effective has not been clearly identified [14,46,47]. According to the review carried out by Silva et al., although exercise-based interventions have a positive impact on the physical functioning and wellbeing of older people, the most effective exercise program in this population remains unidentified [48]. Crocker also indicates that the physical rehabilitation of residents in nursing homes can be effective; however, there is no evidence regarding improvements in sustainability, cost-effectiveness, or which interventions are the most appropriate [49]. Regarding our study, we have shown that, after 12 weeks of exercise training, both the group with basic exercises and functional exercises, with or without verbal stimulation, have a positive effect on improving the physical fitness and quality of life of older people. Other authors have also confirmed that, regardless of the exercise program used, 12 weeks of physical training has a beneficial effect on the functional state of older people [50]. However, de Vreede et al. indicated that the beneficial effect of exercise is lost after suspending activity [51].

Taking everything into consideration, the implemented functional exercise program with verbal stimulation turned out to be the most effective with respect to the short-term effects of the intervention. The greatest positive changes were noted in the performance of complex daily activities, functional fitness of the lower limbs, gait speed and quality of life in the physical domain. In the FET + VS group, after 24 weeks, 1.31 points of improvement were obtained in the SPPB test; 4.13 s improvement in the TUG test; and 7.31 points in the BBS test. These values are higher than the suggested minimum clinically important difference for these tests [52,53].

Providing residents of nursing homes with group physical interventions is profitable and safe. It affects the reduction of disability—recording rare adverse events [54]. As for the residents of nursing homes, maintaining an adequate level of physical and psychological functions, enabling them to perform basic and complex everyday activities, allows them to have control over their own lives. Complete dependence on the help of others is the cause of emotional suffering and feelings of helplessness. According to Prat and Scheicher [55], the loss of functional independence is one of the main problems of older people, while independence increases their satisfaction and improves their quality of life. Weeks et al. [56] stated that preventing the expansion of functional limitations is a key factor motivating older people to participate in physical exercise. An additional element affecting the functioning of older people in nursing homes is the ensemble nature of the institution. In order to improve interactions with other residents, collective training should be used to strengthen social relationships and help to maintain interpersonal harmony, which is necessary for a peaceful life in the facility.

Our study has some limitations. First, there was no research examination performed 36 weeks after the start of the intervention due to the high drop-out rate caused by the increased incidence of influenza among older residents in the nursing homes. Secondly, the collective nature of the facilities means that it was not possible to implement double-blinding. In the case of further studies, additional measurement points should be added after six and 12 months from the beginning of the intervention.

## 5. Conclusions

In summary, the short-term evaluation showed that a functional exercise program, combined with verbal stimulation, is effective in improving physical fitness and raising the level of physical activity spent in free time. To accomplish a sustained functional efficiency and PA change, a prolonged follow-up is required. Finally, the group exercise program is safe and can be implemented into routine practice. Therefore, nursing home staff, as well as relatives, should be involved in the development and implementation of changes aimed at reducing the time spent passively by older people in institutional care.

## Figures and Tables

**Figure 1 jcm-09-00477-f001:**
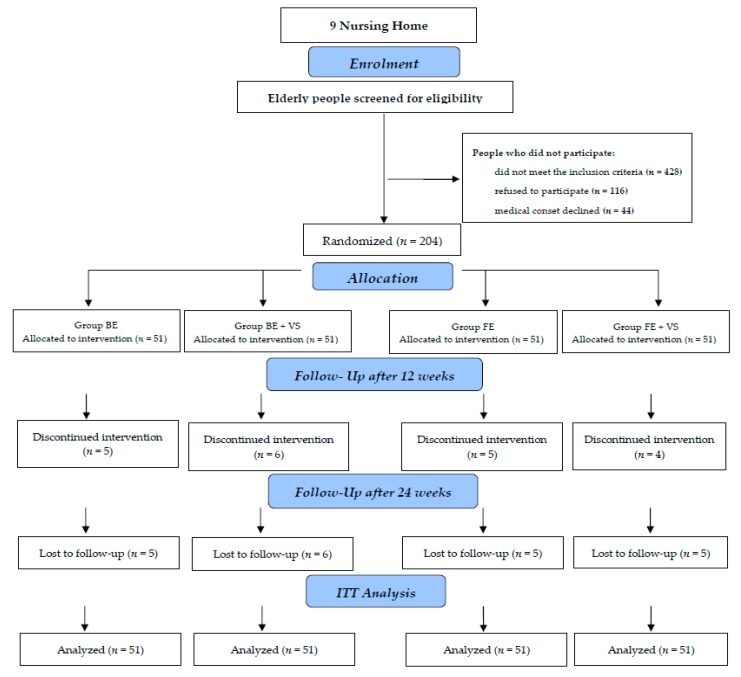
Flow diagram of the intervention study. ITT, intention-to-treat.

**Table 1 jcm-09-00477-t001:** Sociodemographic and clinical characteristics of the participants.

		BE (*n* = 51)	BE + VS (*n* = 51)	FET (*n* = 51)	FET+VS (*n* = 51)	*p*-Value
		Number (%) Mean (SD)				
**Sociodemographic**						
**Sex**	Female	32 (62.75)	29 (56.86)	29 (56.86)	28 (54.90)	*p* = 0.868
	Male	19 (37.25)	22 (43.14)	22 (43.14)	23 (45.10)	
**BMI (kg/m^2^)**	Underweight	1 (1.96)	3 (5.88)	0 (0.00)	1 (1.96)	*p* = 0.266
	Normal body weight	22 (43.14)	20 (39.22)	19 (37.25)	20 (39.22)	
	Overweight	19 (37.25)	11 (21.57)	18 (35.29)	21 (41.18)	
	Obesity	9 (17.65)	17 (33.33)	14 (27.45)	9 (17.65)	
**Marital status**	Married	6 (11.76)	6 (11.76)	6 (11.76)	3 (5.88)	*p* = 0.570
	Widow/widower	21 (41.18)	13 (25.49)	22 (43.14)	22 (43.14)	
	Divorced	7 (13.73)	6 (11.76)	5 (9.80)	8 (15.69)	
	Single	17 (33.33)	26 (50.98)	18 (35.29)	18 (35.29)	
**Education**	Basic	23 (45.10)	16 (31.37)	18 (35.29)	21 (41.18)	*p* = 0.411
	Vocational	11 (21.57)	22 (43.14)	17 (33.33)	15 (29.41)	
	Secondary	14 (27.45)	13 (25.49)	15 (29.41)	14 (27.45)	
	Higher	3 (5.88)	0 (0.00)	1 (1.96)	1 (1.96)	
**Chronic disease**	Cardiovascular	45 (88.24)	42 (82.35)	47 (92.16)	42 (82.35)	*p* = 0.395
	Musculoskeletal	31 (60.78)	32 (62.75)	33 (64.71)	28 (54.90)	*p* = 0.765
	Neurological	10 (19.61)	11 (21.57)	16 (31.37)	9 (17.65)	*p* = 0.354
	Pulmonary	25 (49.02)	24 (47.06)	25 (49.02)	27 (52.94)	*p* = 0.946
	Urinary system	6 (11.76)	8 (15.69)	8 (15.69)	14 (27.45)	*p* = 0.183
**GDS**	No depression	36 (70.59)	33 (64.71)	35 (68.63)	34 (66.67)	*p* = 0.930
	Moderate depression	15 (29.41)	18 (35.29)	16 (31.37)	17 (33.33)	
**MMSE**	No cognitive impairment	16 (31.37)	22 (43.14)	18 (35.29)	17 (33.33)	*p* = 0.805
	Cognitive impairment without dementia	13 (25.49)	12 (23.53)	14 (27.45)	17 (33.33)	
	Mild dementia	22 (43.14)	17 (33.33)	19 (37.25)	17 (33.33)	
	Age (years)	74.37 (8.36)	73.22 (7.33)	74.88 (7.54)	73.76 (7.58)	*p* = 0.717
	Body mass (kg)	70.59 (16.5)	71.81 (20.74)	71.81 (16.15)	71.56 (16.00)	*p* = 0.982
	Height (cm)	163.2 (10.11)	163.24 (11.61)	161.82 (11.19)	164.67 (9.02)	*p* = 0.603
	Years in nursing home	4.18 (3.51)	4.43 (4.11)	3.92 (4.66)	4.10 (3.90)	*p* = 0.937
	Number of drugs	4.16 (1.50)	4.27 (1.73)	4.27 (1.56)	4.25 (1.49)	*p* = 0.978
	Number of falls	0.71 (1.27)	0.88 (1.14)	0.88 (1.42)	0.76 (1.27)	*p* = 0.867
**Main Outcome**						
	SPPB	9.00 (2.20)	9.25 (2.06)	9.22 (2.09)	9.04 (2.24)	*p* = 0.912
**Secondary Outcomes**						
**Physical Activity Assessment**	Total PASE	23.35 (13.61)	23.91 (13.56)	22.92 (13.69)	23.49 (13.07)	*p* = 0.987
	Leisure time activity	6.19 (3.68)	6.26 (3.42)	6.25 (3.68)	6.33 (3.31)	*p* = 0.998
	Household activity	17.16 (11.72)	17.65 (11.50)	16.67 (11.9)	17.16 (11.72)	*p* = 0.981
**Functional Assessment**	ADL	5.29 (0.86)	5.18 (0.89)	5.25 (0.93)	5.35 (0.84)	*p* = 0.782
	IADL	8.61 (2.07)	8.59 (2.5)	8.67 (2.45)	8.78 (2.28)	*p* = 0.974
**Muscular Strength Assessments**	HGS _P_ (kg)	18 (8.77)	19.12 (9.77)	19.96 (10.88)	18.65 (9.04)	*p* = 0.773
	HGS _L_ (kg)	15.82 (9.25)	17.99 (9.06)	17.03 (9.03)	17.93 (8.71)	*p* = 0.584
	5× STS (s)	22.83 (10.54)	23.87 (8.5)	23.80 (10.90)	23.96 (11.17)	*p* = 0.940
**Mobility Assessment**	TUG (s)	21.07 (9.88)	20.2 (9.87)	21.32 (10.72)	20.99 (10.72)	*p* = 0.952
	TUG _cog_ (s)	25.17 (11.58)	24.33 (11.19)	25.00 (11.08)	24.88 (12.25)	*p* = 0.985
	Gait speed (m/s)	0.59 (0.27)	0.62 (0.32)	0.57 (0.27)	0.60 (0.27)	*p* = 0.828
**Flexibility Assessment**	BS _R_ (cm)	−32.82 (16.74)	−31.59 (16.31)	−33.57 (13.46)	−31.39 (13.40)	*p* = 0.868
	BS _L_ (cm)	−34.18 (17.41)	−34.22 (16.79)	−34.9 (13.41)	−31.20 (18.47)	*p* = 0.681
	CSR _R_ (cm)	−14.06 (14.8)	−13.98 (16.04)	−11.29 (12.69)	−11.59 (13.82)	*p* = 0.649
	CSR _L_ (cm)	−14.59 (15.2)	−14.88 (15.21)	−11.59 (12.5)	−12.53 (13.83)	*p* = 0.589
**Body Balance Assessment**	BBS	33.96 (13.15)	33.10 (13.20)	32.67 (14.40)	33.00 (13.94)	*p* = 0.969
**Other Outcomes**						
**Quality of Life Assessment**	Physical Component Summary	52.01 (15.74)	51.61 (14.80)	54.06 (15.44)	54.51 (3.34)	*p* = 0.693
	Mental Health Component Summary	58.09 (15.42)	59.24 (19.07)	61.48 (14.71)	62.75 (14.89)	*p* = 0.456

SD, Standard Deviation; BMI, body mass index; GDS, Geriatric Depression Scale; MMSE, Mini-Mental State Examination; SPPB, Short Physical Performance Battery; PASE, Physical Activity for Elderly; ADL, Activity of Daily Living; IADL, Instrumental Activity of Daily Living; HGS, Handgrip Strength; 5× STS, 5× Sit to Stand; TUG, Timed Up and Go; TUG cog, Timed Up and Go cognitive; BS, Back Stretch; CSR, Chair Sit and Reach; BBS, Berg Balance Scale.

**Table 2 jcm-09-00477-t002:** Mean difference scores for each group across time.

		BE	BE + VS	FET	FET+ VS	BE	BE + VS	FE	FE + VS
		Baseline—12 weeks				Baseline—24 weeks			
**Main Outcome**		Mean change from baseline (95% CI)							
	SPPB	0.25	0.51	0.73	1.27	−0.04	0.41	0.53	1.31
		(0.10–0.41) *	(0.27–0.75) *	(0.47–0.98) *	(0.88–1.67) *	(−0.23–0.15)	(0.17–0.66) *	(0.28–0.78) *	(0.93–1.70) *
**Secondary Outcomes**									
**Physical Activity Assessment**	Total PASE	5.50	4.83	4.02	5.46	1.44	5.59	4.14	6.91
		(4.85–6.14) *	(3.33–6.32) *	(1.78–6.27) *	(4.89–6.02) *	(0.70–2.19) *	(2.98–8.20) *	(0.94–7.34) *	(4.58–9.24) *
	Leisure time activity	5.50	5.42	5.40	5.50	1.44	3.73	2.57	5.77
		(4.85–6.14) *	(4.84–5.99) *	(4.84–5.95) *	(4.98–6.02) *	(0.70–2.19) *	(2.77–4.68) *	(1.64–3.51) *	(4.32–7.22) *
	Household activity	0.00	0.00	0.00	0.00	0.00	2.45	2.94	1.18
		(0.00–0.00)	(0.00–0.00)	(0.00–0.00)	(0.00–0.00)	(0.00–0.00)	(0.34–4.56) *	(0.65–5.23) *	(−0.50–2.85)
**Functional Assessment**	ADL	0.06	0.04	0.25	0.31	−0.16	−0.08	0.10	0.29
		(−0.07–0.19)	(−0.13–0.21)	(0.10–0.41) *	(0.13–0.50) *	(−0.27–−0.04) *	(−0.20–0.05)	(0.01–0.18) *	(0.09–0.50) *
	IADL	0.73	0.80	0.94	1.51	0.16	0.37	0.53	1.33
		(0.34–1.12) *	(0.39–1.22) *	(0.52–1.36) *	(1.08–1.94) *	(0.03–0.29) *	(0.13–0.62) *	(0.24–0.82) *	(0.87–1.80) *
**Muscular Strength Assessments**	HGS _P_ (kg)	0.65	2.51	3.08	3.63	−0.13	2.14	2.07	3.73
		(−0.30–1.60)	(1.18–3.83) *	(0.98–5.18) *	(2.21–5.06) *	(−1.08–0.81)	(0.86–3.42) *	(0.43–3.70) *	(2.51–4.95) *
	HGS _L_ (kg)	0.71	1.77	2.85	3.27	0.16	1.10	2.17	3.41
		(−0.04–1.46)	(0.63–2.91) *	(1.29–4.40) *	(1.92–4.63) *	(−0.87–1.18)	(0.02–2.19) *	(0.58–3.76) *	(2.20–4.62) *
	5× STS (s)	−2.55	−3.98	−4.46	−6.36	−0.73	−2.86	−3.75	−6.30
		(−3.80–−1.30) *	(−5.72–−2.23) *	(−6.20–−2.71) *	(−8.43–−4.29) *	(−1.83–0.36)	(−4.35–−1.38) *	(−5.47–−2.03) *	(−8.40–−4.19) *
**Mobility Assessment**	TUG (s)	−0.91	−1.63	−3.32	−3.89	0.25	−0.24	−0.71	−4.13
		(−1.60–−0.23) *	(−2.93–−0.33) *	(−5.16–−1.48) *	(−5.82–−1.97) *	(−0.72–1.23)	(−1.37–0.88)	(−2.04–0.62)	(−6.09–−2.18) *
	TUG _cog_ (s)	−1.29	−1.85	−2.79	−3.61	0.08	−0.10	−0.70	−3.88
		(−2.42–−0.16) *	(−3.18–−0.51) *	(−3.87–−1.72) *	(−5.87–−1.36) *	(−1.30–1.47)	(−1.45–1.26)	(−2.21–0.80)	(−6.66–−1.10) *
	Gait speed (m/s)	0.07	0.09	0.09	0.17	−0.02	0.01	0.04	0.17
		(0.04–0.09) *	(0.05–0.13) *	(0.05–0.14) *	(0.13–0.21) *	(−0.04–0.01)	(−0.02–0.05)	(0.01–0.07) *	(0.13–0.22) *
**Flexibility Assessment**	BS _R_ (cm)	1.57	7.16	6.41	9.27	0.82	5.14	5.31	9.24
		(−0.74–3.88)	(4.49–9.82) *	(4.76–8.06) *	(5.94–12.61) *	(−1.45–3.10)	(2.53–7.75) *	(3.20–7.42) *	(5.82–12.65) *
	BS _L_ (cm)	1.63	7.18	8.31	7.53	0.59	4.49	6.41	8.80
		(−0.44–3.70)	(4.57–9.78) *	(5.67–10.96) *	(3.00–12.06) *	(−1.49–2.66)	(1.98–7.00) *	(4.24–8.58) *	(4.01–13.59) *
	CSR _R_ (cm)	4.16	7.90	10.02	11.43	2.22	5.55	6.78	10.86
		(2.18–6.14) *	(3.96–11.85) *	(6.94–13.10) *	(7.96–14.90) *	(0.29–4.14) *	(2.16–8.94) *	(3.93–9.64) *	(7.67–14.06) *
	CSR _L_ (cm)	3.90	8.76	10.73	10.94	2.10	6.80	8.41	10.84
		(1.52–6.29) *	(6.18–11.35) *	(7.75–13.70) *	(7.99–13.89) *	(0.49–3.70) *	(4.21–9.40) *	(5.60–11.23) *	(7.82–13.87) *
**Body Balance Assessment**	BBS	1.18	4.27	5.82	7.27	0.20	2.78	4.39	7.31
		(−0.47–2.83)	(2.82–5.73) *	(3.65–8.00) *	(5.27–9.28) *	(−1.64–2.03)	(1.67–3.90) *	(2.29–6.49) *	(5.40–9.23) *
**Other Outcomes**									
**Quality of Life Assessment**	Physical Component Summary	0.97	6.46	5.46	12.10	−0.21	4.80	2.78	11.79
		(−1.94–3.87)	(3.24–9.67) *	(3.00–7.92) *	(9.33–14.86) *	(−2.22–1.80)	(2.04–7.56) *	(0.23–5.32) *	(8.64–14.95) *
	Mental Health Component Summary	2.28	7.88	5.18	7.28	0.04	6.34	4.27	5.01
		(−1.44–5.99)	(3.30–12.45) *	(2.35–8.01) *	(3.50–11.07) *	(−2.43–2.50)	(2.04–10.63) *	(1.55–6.99) *	(1.22–8.79) *

* statistically significant result. SPPB, Short Physical Performance Battery; PASE, Physical Activity for Elderly; ADL, Activity of Daily Living; IADL, Instrumental Activity of Daily Living; HGS, Handgrip Strength; 5× STS, 5× Sit to Stand; TUG, Timed Up and Go; TUG cog, Timed Up and Go cognitive; BS, Back Stretch; CSR, Chair Sit and Reach; BBS, Berg Balance Scale.

**Table 3 jcm-09-00477-t003:** Between-group comparisons at 12 weeks.

		Post hoc (Bonferroni) Analysis						ANOVA *p* Value
		BE vs. BE + VS	BE vs. FET	BE vs. FET + VS	FET vs. BE + VS	BE + VS vs. FET + VS	FET vs. FET+ VS	
**Main Outcome**								
	SPPB	*p* = 0.387	*p* = 0.051	*p* < 0.001 *	*p* = 0.387	*p* = 0.001 *	*p* = 0.022 *	*p* < 0.001 *
**Secondary Outcomes**								
**Functional Assessment**	ADL	*p* = 1.000	*p* = 0.255	*p* = 0.127	*p* = 0.233	*p* = 0.097	*p* = 1.000	*p* = 0.033 *
	IADL	*p* = 1.000	*p* = 1.000	*p* = 0.047 *	*p* = 1.000	*p* = 0.082	*p* = 0.210	*p* = 0.034 *
**Muscular Strength Assessments**	HGS _P_ (kg)	*p* = 0.330	*p* = 0.116	*p* = 0.033 *	*p* = 1.000	*p* = 0.871	*p* = 1.000	*p* = 0.033 *
	HGS _L_ (kg)	*p* = 0.657	*p* = 0.075	*p* = 0.022 *	*p* = 0.657	*p* = 0.346	*p* = 0.657	*p* = 0.017 *
	5× STS (s)	*p* = 0.483	*p* = 0.474	*p* = 0.012 *	*p* = 0.695	*p* = 0.257	*p* = 0.474	*p* = 0.020 *
**Mobility Assessment**	TUG (s)	*p* = 1.000	*p* = 0.129	*p* = 0.036 *	*p* = 0.349	*p* = 0.144	*p* = 1.000	*p* = 0.018 *
	TUG _cog_ (s)	*p* = 1.000	*p* = 0.655	*p* = 0.191	*p* = 1.000	*p* = 0.509	*p =* 1.000	*p =* 0.144
	Gait speed (m/s)	*p* = 0.935	*p* = 0.935	*p* = 0.001 *	*p* = 0.935	*p* = 0.021 *	*p* = 0.027 *	*p =* 0.002 *
**Flexibility Assessment**	BS _R_ (cm)	*p* = 0.011 *	*p* = 0.032 *	*p* < 0.001 *	*p* = 0.680	*p* = 0.484	*p* = 0.343	*p* < 0.001 *
	BS _L_ (cm)	*p* = 0.048 *	*p* = 0.015 *	*p* = 0.038 *	*p* = 1.000	*p* = 1.000	*p* = 1.000	*p* = 0.010 *
	CSR _R_ (cm)	*p* = 0.393	*p* = 0.050 *	*p* = 0.009 *	*p* = 0.697	*p* = 0.393	*p* = 0.697	*p* = 0.009 *
	CSR _L_ (cm)	*p* = 0.050 *	*p* = 0.002 *	*p* = 0.002 *	*p* = 0.780	*p* = 0.780	*p* = 0.911	*p* = 0.001 *
**Body Balance Assessment**	BBS	*p* = 0.071	*p* = 0.002 *	*p* < 0.001 *	*p* = 0.468	*p* = 0.071	*p* = 0.468	*p* < 0.001 *
**Other Outcomes**								
**Quality of Life Assessment**	Physical Component Summary	*p* = 0.022 *	*p* = 0.052	*p* < 0.001 *	*p* = 0.620	*p* = 0.022 *	*p* = 0.006 *	*p* < 0.001 *
	Mental Health Component Summary	*p* = 0.218	*p* = 1.000	*p* = 0.306	*p* = 1.000	*p* = 1.000	*p* = 1.000	*p* = 0.146

* statistically significant result. SPPB, Short Physical Performance Battery; ADL, Activity of Daily Living; IADL, Instrumental Activity of Daily Living; HGS, Handgrip Strength; 5× STS, 5× Sit to * statistically significant result. SPPB, Short Physical Performance Battery; ADL, Activity of Daily Living; IADL, Instrumental Activity of Daily Living; HGS, Handgrip Strength; 5× STS, 5× Sit to Stand; TUG, Timed Up and Go; TUG cog, Timed Up and Go cognitive; BS, Back Stretch; CSR, Chair Sit and Reach; BBS, Berg Balance Scale.

**Table 4 jcm-09-00477-t004:** Between-group comparisons at 24 weeks.

		Post hoc (Bonferroni) Analysis						ANOVA *p* Value
		BE vs. BE + VS	BE vs. FET	BE vs. FET + VS	FET vs. BE + VS	BE + VS vs. FET + VS	FET vs. FET + VS	
**Main Outcome**								
	SPPB	*p* = 0.044 *	*p* = 0.012 *	*p* < 0.001 *	*p* = 0.547	*p* < 0.001 *	*p* < 0.001 *	*p* < 0.001 *
**Secondary Outcomes**								
**Physical Activity Assessment**	Total PASE	*p* = 0.075	*p* = 0.413	*p* = 0.009 *	*p* = 0.787	p = 0.787	*p* = 0.413	*p* = 0.010 *
	Leisure time activity	*p* = 0.010 *	*p* = 0.245	*p* <0.001 *	*p* = 0.245	*p* = 0.019 *	*p* < 0.001 *	*p* < 0.001 *
	Household activity	*p* = 0.252	*p* = 0.115	*p* = 0.922	*p* = 0.922	*p* = 0.922	*p* = 0.632	*p* = 0.083
**Functional Assessment**	ADL	*p* = 0.427	*p* = 0.042 *	*p* <0.001 *	*p* = 0.150	*p* = 0.001 *	*p* = 0.144	*p* < 0.001 *
	IADL	*p* = 0.640	*p* = 0.260	*p* <0.001 *	*p* = 0.640	p<0.001 *	*p* = 0.001 *	*p* < 0.001 *
**Muscular Strength Assessments**	HGS _P_ (kg)	*p* = 0.066	*p* = 0.066	*p* <0.001 *	*p* = 0.933	*p* = 0.208	*p* = 0.208	*p* = 0.001 *
	HGS _L_ (kg)	*p* = 0.479	*p* = 0.092	*p* = 0.002 *	*p* = 0.479	*p* = 0.047 *	*p* = 0.479	*p* = 0.002 *
	5× STS (s)	*p* = 0.134	*p* = 0.039 *	*p* < 0.001 *	*p* = 0.444	*p* = 0.017 *	*p* = 0.087	*p* < 0.001 *
**Mobility Assessment**	TUG (s)	*p* = 1.000	*p* = 0.989	*p* < 0.001 *	*p* = 1.000	*p* = 0.001 *	*p* = 0.002 *	*p* < 0.001 *
	TUG _cog_ (s)	*p* = 1.000	*p* = 1.000	*p* = 0.016 *	*p* = 1.000	*p* = 0.021 *	*p* = 0.063	*p* = 0.008 *
	Gait speed (m/s)	*p* = 0.366	*p* = 0.045 *	*p* < 0.001 *	*p* = 0.366	*p* < 0.001 *	*p* <0.001 *	*p* < 0.001 *
**Flexibility Assessment**	BS _R_ (cm)	*p* = 0.087	*p* = 0.085	*p*< 0.001 *	*p* = 0.925	*p* = 0.088	*p* = 0.088	*p* < 0.001 *
	BS _L_ (cm)	*p* = 0.224	*p* = 0.040 *	*p* = 0.001 *	*p* = 0.546	*p* = 0.196	*p* = 0.546	*p* = 0.002 *
	CSR _R_ (cm)	*p* = 0.208	*p* = 0.105	*p* < 0.001 *	*p* = 0.546	*p* = 0.049 *	*p* = 0.141	*p* = 0.001 *
	CSR _L_ (cm)	*p* = 0.04 *	*p* = 0.003 *	*p* < 0.001 *	*p* = 0.375	*p* = 0.080	*p* = 0.360	*p* < 0.001 *
**Body Balance Assessment**	BBS	*p* = 0.080	*p* = 0.004 *	*p* < 0.001 *	*p* = 0.201	*p* = 0.002 *	*p* = 0.062	*p* <0.001 *
**Other Outcomes**								
**Quality of Life Assessment**	Physical Component Summary	*p* = 0.024 *	*p* = 0.222	*p* < 0.001 *	*p* = 0.280	*p* = 0.001 *	*p* <0.001 *	*p* <0.001 *
	Mental Health Component Summary	*p* = 0.055	*p* = 0.313	*p* = 0.195	*p* = 1.000	*p* = 1.000	*p =* 1.000	*p* = 0.054

* statistically significant result. SPPB, Short Physical Performance Battery; PASE, Physical Activity for Elderly; ADL, Activity of Daily Living; IADL, Instrumental Activity of Daily Living; HGS, Handgrip Strength; 5× STS, 5× Sit to Stand; TUG, Timed Up and Go; TUG cog, Timed Up and Go cognitive; BS, Back Stretch; CSR, Chair Sit and Reach; BBS, Berg Balance Scale.

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
