# Peer review of "Effects of Physical Exercises and Verbal Stimulation on the Functional Efficiency and Use of Free Time in an Older Population under Institutional Care: A Randomized Controlled Trial"

_jcm, 2020, doi:10.3390/jcm9020477_

Round 1

Reviewer 1 Report

As an overarching comment, the authors should carefully follow and report the information described in the trial registry and published protocol paper for this specific trial. Then, if it occurs any changes or difficulties during data collection or analytic processes, the authors carefully explain the reasons for the changes or inconsistencies in the manuscript. The current version of your manuscript still contains lots of inconsistencies. This seriously violates the trial integrity which is more than important as phase IV (as the authors described so in the trial registration) confirmatory RCT. As you know, the phase IV trial is considered a final step to verify the effectiveness of an intervention and possibly leads to application in clinical practice. Hence, strict reporting manner must be followed. If the authors violate this, the trial should be reported as exploratory not confirmatory (phase IV).

According to the descriptions in the author’s protocol paper for this trial (https://doi.org/10.1186/s13063-017-2114-1), data should be analyzed on an intention-to-treat basis. The missing data should be handled by multiple imputation method.

“The analysis will be performed on the ITT set. The analysis will be repeated per-protocol (PP) (in which participants who violate the protocol are excluded from the analysis set) and compared to the ITT analysis as a sensitivity analysis. The multiple imputation (MI) technique will used for dealing with missing data under the assumption that data are missing at random [53].”

Again, this reviewer still does not understand the rationales for choosing Cohen’s effect size of 0.5 without any supporting preliminary/pilot data. Although it may be ‘standard’ effect size, 0.5 would be quite large when having an active control group. The authors should explain why they determined Cohen’s ES of 0.5.

Line 24. ‘In the studied groups, the elderly did not differ from each other.’ This sentence does not make sense. It should be replaced by another expression to convey what the authors want to say.

Lines 28-29. Changes is SPPB score, gait speed, grip strength, and flexibility should be accompanied with 95% confidence intervals as well as each point estimates.

Table 3. Baseline-week 12 for PASE Household may be incorrect data? All cells are zeros.

The authors mention that this research received no external funding. In my opinion, this type of RCT is obviously impossible to be implemented without funding support. Federal, regional or institutional funding might have been obtained. If so the authors should clearly report them.

This reviewer highly recommends, maybe again, making this manuscript more simple and concise, and easily understandable to readership of the journal. The current form of manuscript looks too long and contains overwhelming amount or size of tables. These are almost impossible to follow. 

Author Response

Response to Reviewer

We would like to thank for the comments and suggestions which offered an opportunity for further improvement of our manuscript.

The manuscript has been carefully rechecked and appropriate changes have been made in accordance with the reviewers’ comments. Changes to the main text of manuscript according to these comments have been marked in red font. A native English speaker checked the manuscript for linguistic errors and clarity, any minor edits in language are not highlighted in the text.

Once again, we thank the reviewers for thoughtful suggestions and insights, which have enriched our manuscript and produced a more balanced and better account of our research. We thank the reviewers for their constructive criticism and kindly ask to consider this revised version of the manuscript for publication in Journal of Clinical Medicine. 

Point 1: As an overarching comment, the authors should carefully follow and report the information described in the trial registry and published protocol paper for this specific trial. Then, if it occurs any changes or difficulties during data collection or analytic processes, the authors carefully explain the reasons for the changes or inconsistencies in the manuscript. The current version of your manuscript still contains lots of inconsistencies. This seriously violates the trial integrity which is more than important as phase IV (as the authors described so in the trial registration) confirmatory RCT. As you know, the phase IV trial is considered a final step to verify the effectiveness of an intervention and possibly leads to application in clinical practice. Hence, strict reporting manner must be followed. If the authors violate this, the trial should be reported as exploratory not confirmatory (phase IV).

Response 1: We thank the reviewer for this comment. We agree that randomized trials with long-term follow-up can provide estimates of the long-term effects of health interventions. However, our analysis of long-term outcomes was complicated by problems with outcome measurements arising from loss to follow-up, diseases and death of participants.

As a result, we have added the additional sentences to our conclusion:

“In summary, the short-term evaluation showed that a functional exercise program combined with verbal stimulation is effective in improving physical fitness and raising the level of physical activity spent in free time. To accomplish a sustained functional efficiency and PA change, a prolonged follow-up is required. Finally, the group exercise program is safe and can be implemented into routine practice, therefore, nursing home staff as well as relatives should be involved in the development and implementation of changes aimed at reducing the time spent passively by older people under institutional care.”       

Point 2: According to the descriptions in the author’s protocol paper for this trial (https://doi.org/10.1186/s13063-017-2114-1), data should be analyzed on an intention-to-treat basis. The missing data should be handled by multiple imputation method.

“The analysis will be performed on the ITT set. The analysis will be repeated per-protocol (PP) (in which participants who violate the protocol are excluded from the analysis set) and compared to the ITT analysis as a sensitivity analysis. The multiple imputation (MI) technique will used for dealing with missing data under the assumption that data are missing at random [53].”

Response 2: In response to the reviewer’s comment, we have added analysis performed on the ITT set. We changed Figure 1 and Tables 2,3 and 4.

We have added the following information to description 2.7 section:

“Descriptive characteristics were presented as mean and standard deviation or number and percent when appropriate. A one-way ANOVA test was used to assess differences between groups. The mean difference between treatment groups and the confidence intervals for quantitative variables was also determined. Post-hoc analysis for quantitative variables analysis was conducted with t-tests with Bonferroni correction. Significance of changes in examined variables between two time points were assessed with paired t-test. Standard intention-to-treat analysis was performed for each outcome. Missing data were deemed to be missing at random and calculated using the imputation technique according to the protocol study [37]. Analyses were conducted at 0.05 level of significance. R software, version 3.6.1 was used. “

Point 3: Again, this reviewer still does not understand the rationales for choosing Cohen’s effect size of 0.5 without any supporting preliminary/pilot data. Although it may be ‘standard’ effect size, 0.5 would be quite large when having an active control group. The authors should explain why they determined Cohen’s ES of 0.5.

Response 3: Before starting our study, we calculated the power of study with an estimated effect size. For the calculating a reasonable sample size, the changes in SPPB scores from previous research was considered by our biostatistician. When we planned the study, in an a-priori power analysis, a biostatistician calculated the effect size based on Cohen’s [1].

Cohen, J. (1992). A power primer. Psychol Bull.

Point 4: Line 24. ‘In the studied groups, the elderly did not differ from each other.’ This sentence does not make sense. It should be replaced by another expression to convey what the authors want to say.

Response 4: We are very sorry for this inaccuracy.

The sentence, “In the studied groups, the elderly did not differ from each other “has been changed to, “No statistically significant differences in baseline characteristics were observed across the groups.”

Point 5: Lines 28-29. Changes is SPPB score, gait speed, grip strength, and flexibility should be accompanied with 95% confidence intervals as well as each point estimates.

Response 5: We have added 95% CI to the description of results in abstract:

 “In the group with functional exercise training with verbal stimulation compared to the group with basic exercises, the greatest positive short-term impact of intervention was demonstrated in terms of functional fitness (SPPB increased by 1.31 points; 95% Confidence Interval (CI) = 0.93;1.70), gait speed (improved by 0.17 m/s, 95% CI = 0.13;0.21 ), hand grip strength (by over 4 kg; 95% CI = 2.51;4.95), and upper limb flexibility (by 10 cm; 95% CI = 5.82;12.65). There was also a significant increase in the level of free time physical activity and an improvement in the quality of life especially expressed in the domain of overall physical functioning.”

Point 6: Table 3. Baseline-week 12 for PASE Household may be incorrect data? All cells are zeros.

Response 6: The results are recorded correctly. All participants had the same level of involvement in homework during I and II assessment.

Point 7: The authors mention that this research received no external funding. In my opinion, this type of RCT is obviously impossible to be implemented without funding support. Federal, regional or institutional funding might have been obtained. If so the authors should clearly report them.

Response 7: The research was conducted as a part of our full-time employment at the University of Rzeszow, and we did not receive a grant from another institution. The University of Rzeszow, with its own funds for statutory research projects, covered the costs of field research (as part of so-called own research projects - such internal projects do not receive any numbers / reference numbers in that case) We are aware of the fact that the study required large financial outlays, however, most of the work was based on the voluntary support of employees of nursing homes, whose directors agreed to take part in the project.

We have added the following information to the manuscript:

“Funding: This study was supported by the funds for statutory research of the University of Rzeszow”

Point 8: This reviewer highly recommends, maybe again, making this manuscript simpler and more concise, and easily understandable to readership of the journal. The current form of manuscript looks too long and contains overwhelming amount or size of tables. These are almost impossible to follow.

Response 8: We are very grateful for your insightful comments. We have made every effort to improve our manuscript, as reviewer’s suggestions.  We have shortened the manuscript text and reduced the number of tables. We hope that the current manuscript form will meet Reviewers expectations.

Reviewer 2 Report

Thanks for the opportunity to review this paper. I think the study followed a very well-designed methodology. While I do not have any major concerns, I have a few questions/comments for the authors. I hope the comments are helpful.

The introduction section seems to be too exhaustive - in some places, I felt like the authors dragged the background of the study too much. Could the authors please revisit their Introduction section to make it concise and more focused. Throughout the manuscript, the authors used "physical activity" and "physical exercise" synonymously, which is incorrect. This paper (https://www.ncbi.nlm.nih.gov/pmc/articles/PMC1424733/) explains the differences between these terms. The authors should consider using the appropriate terminology throughout the manuscript.  One thing that I am not clear from the description of the methods (section 2.1, Trial design) is whether or not data from the participants in different groups collected at the same time. It took ~2 and a half years to recruit the participants, did the authors recruit and collect data at the same time - can the authors please clarify why?      Section 2.3: Was there any reason for not to have a control arm? I understand who assessed outcome measures did not take part in intervention delivery. Were those who analysed the data were blinded or independent to assessment/implementation?  Table 2: Can the authors please clarify how the group differences were measured for categorical variables? Chi2 test, exact test?  Those who discontinued intervention, were they significantly different from the those who completed? How lost to follow-up could have affected the results? Please clarify if an intention-to-treat analysis was applied.   

Author Response

Response to Reviewer

We would like to thank for the comments and suggestions which offered an opportunity for further improvement of our manuscript.

The manuscript has been carefully rechecked and appropriate changes have been made in accordance with the reviewers’ comments. Changes to the main text of manuscript according to these comments have been marked in red font. A native English speaker checked the manuscript for linguistic errors and clarity, any minor edits in language are not highlighted in the text.

Once again, we thank the reviewers for thoughtful suggestions and insights, which have enriched our manuscript and produced a more balanced and better account of our research. We thank the reviewers for their constructive criticism and kindly ask to consider this revised version of the manuscript for publication in Journal of Clinical Medicine. 

Point 1: Thanks for the opportunity to review this paper. I think the study followed a very well-designed methodology. While I do not have any major concerns, I have a few questions/comments for the authors. I hope the comments are helpful.

The introduction section seems to be too exhaustive - in some places, I felt like the authors dragged the background of the study too much. Could the authors please revisit their Introduction section to make it concise and more focused. Throughout the manuscript, the authors used "physical activity" and "physical exercise" synonymously, which is incorrect. This paper (https://www.ncbi.nlm.nih.gov/pmc/articles/PMC1424733/) explains the differences between these terms. The authors should consider using the appropriate terminology throughout the manuscript. 

Response 1: We thank the reviewer for this comment as, as a result, we have shortened the introduction section. We tried to use the appropriate terminology. We thoroughly reviewed the terms we used for physical activity and physical exercise.    

Point 2: One thing that I am not clear from the description of the methods (section 2.1, Trial design) is whether or not data from the participants in different groups collected at the same time. It took ~2 and a half years to recruit the participants, did the authors recruit and collect data at the same time - can the authors please clarify why?    

Response 2: The same measuring time points were used in each of the study group. Outcome assessment were performed at the start of the study and then after 12 and 24 weeks. The entire study lasted over two and a half years, including: promotional activities, recruitment, preparation of intervention, conducting it and then developing data.

We apologize for the error, which appeared in the description. We added following sentence in the end of section 2.1.

“It was a randomized controlled trial with four open-label, parallel groups.  The study was conducted in the period from March 2016 to December 2018. Data was collected at baseline and at 12- and 24-weeks. This procedure was established according to "CONSORT" statement [26]. The study was registered in the Sri Lanka Clinical Trials Registry (SLCTR / 2016/004).”

We added the following sentence at the beginning of section 2.4.

“Outcome assessments were performed at baseline, 12 and 24-weeks.”

Point 3: Was there any reason for not to have a control arm?

Response 3: Our review of the literature showed that the majority of intervention studies conducted among older people have a "passive" control group. Orrow et al. in systematic review showed significantly larger intervention effects on self-reported activity and functional efficiency in studies where control participants received no intervention than where they received a comparator intervention [1]. However, there are few studies comparing the effectiveness of various interventions. [2].  Therefore, we decided to design four parallel groups with exercises and we think that this design is most ethical for older people under institutional care [3].

Orrow, G., Kinmonth, A. L., Sanderson, S., & Sutton, S. (2012). Effectiveness of physical activity promotion based in primary care: systematic review and meta-analysis of randomised controlled trials. BMJ (Clinical research ed.), 344, e1389. de Vries, N.M., van Ravensberg, C.D., Hobbelen, J.S., Olde Rikkert, M.G., Staal, J.B., Nijhuis-van der Sanden, M.W. (2012). Effects of physical exercise therapy on mobility, physical functioning, physical activity and quality of life in community-dwelling older adults with impaired mobility, physical disability and/or multi-morbidity: a meta-analysis. Ageing Res Rev, 11(1), 136-149. Nair B. (2019). Clinical Trial Designs. Indian dermatology online journal, 10(2), 193–201.

Point 4: I understand who assessed outcome measures did not take part in intervention delivery. Were those who analysed the data were blinded or independent to assessment/implementation? 

Response 4: We mentioned this in section 2.6.

“Outcome assessors were blinded to group division and did not take part in implementing interventions”

Point 5: Can the authors please clarify how the group differences were measured for categorical variables? Chi2 test, exact test?  Those who discontinued intervention, were they significantly different from the those who completed? How lost to follow-up could have affected the results? Please clarify if an intention-to-treat analysis was applied. 

Response 5: We conducted an additional analysis assessing whether people who dropped out of the study disturb the balanced distribution of older people between groups. The analysis showed that the groups studied at individual measuring points did not differ from each other in terms of sociodemographic features.

We made intention-to-treat analysis during review process. All participants who were included in the randomization process were analysed.

We have added the following information to description 2.8 section:

Descriptive characteristics were presented as mean and standard deviation or number and percent when appropriate. A one-way ANOVA test was used to assess differences between groups. The mean difference between treatment groups and the confidence intervals for quantitative variables was also determined. Post-hoc analysis for quantitative variables analysis was conducted with t-tests with Bonferroni correction. Significance of changes in examined variables between two time points were assessed with paired t-test. Standard intention-to-treat analysis was performed for each outcome. Missing data were deemed to be missing at random and calculated using the imputation technique according to the protocol study [37]. Analyses were conducted at 0.05 level of significance. R software, version 3.6.1 was used.

Round 2

Reviewer 1 Report

This reviewer recommends transferring data on postural controls to supplemental files. These are considered "other outcomes", have many detailed variables (eyes closed and opened), and occupy the several pages. Again, I highly recommends making your manuscript as concise as you can though this journal has no page limit.

Author Response

Response to Reviewer

We would like to thank for the comments and suggestions which offered an opportunity for further improvement of our manuscript.

English language editing was undergone by MDPI. The text has been checked for correct use of grammar and common technical terms, and edited to a level suitable for reporting research in a scholarly journal.

We are now resubmitting the revised manuscript and also the point-by-point response to the comments.

Point 1: This reviewer recommends transferring data on postural controls to supplemental files. These are considered "other outcomes", have many detailed variables (eyes closed and opened), and occupy the several pages.

Response 1: We would like to thank for the comments.

We added postural control data to supplemental files.

Supplementary Materials: Table S1: Postural balance characteristics of the participants, Table S2: Mean difference scores for each group across time, Table S3: Between-group comparisons in postural balance at 12 weeks, Table S4: Between-group comparisons in postural balance at 24 weeks.

Point 2: Again, I highly recommends making your manuscript as concise as you can though this journal has no page limit.

Response 2: We have made every effort to improve our manuscript. We tried to make our manuscript more concise and at the same time included all points from the CONSORT checklist. We hope that the revised manuscript will meet reviewer's expectations and we are willing to answer any other questions you might have.

This manuscript is a resubmission of an earlier submission. The following is a list of the peer review reports and author responses from that submission.

Round 1

Reviewer 1 Report

Statistics and paper design are seen as well-organized studies. In recent years, the importance of community and institutional care to the elderly population is increasing. At such a point, it is appropriate research.

As a new treatment method through physical training through verbal stimulation, it is important to demonstrate the effectiveness of exercise that cannot be solved by simple physical training.

In particular, it makes sense that the study demonstrated the effectiveness of exercise in randomized studies.

General comment

Please rewrite with commas and periods.

Please check the abbreviations on the tables and figures again.

Please summarize the table more briefly and move to supplementary table.

Methods

Please add a reference to how you set criteria for gait speed, hand grip strength, and upper limb flexibility.

Reviewer 2 Report

Major comments

Where is data at week 36 (24 weeks after the end of interventions)? According to the trial information registered in the Sri Lanka CTR, this trial seems to collect follow-up data at week 36. Data at week 36 must be more appropriate for evaluating long-term effectiveness.

What is the evidence for the effect size (Cohen' d) of 0.5? Is the value based on any preliminary/pilot trials? If not, how did the authors determine it? Where do the authors want to find between-group difference, BE vs. FET plus VS? Should be pre-specified.

Why did not the authors analyze data based on an intention-to-treat principal?  The ITT is now considered standard way of analysis for the confirmatory RCT and is highly recommended by the CONSORT statement. The authors stated that the manuscript follows the CONSORT reporting guideline. How were the missing data handled?

The authors should clearly pre-specify the primary outcomes and its assessment time frame. These are also recommended by the CONSORT statement. The registered information reported that TUG, 10-meter walk, back scratch test, sit and reach, Berg balance scale, and PASE are primary outcomes of interest. Their time points for determining the trial effectiveness are not specified in the trial registration. Assigning multiple primary outcomes (six outcomes in your trial) places your trial at the high risk of type I errors due to multiplicity (i.e., many outcomes are tested at multiple time points).

If the trial does not follow the ITT approach, the authors should adjust potential confounding variables to test primary effectiveness. This is because deleting data from some participants can violate the assumption that background characteristics are completely/theoretically balanced between groups.

Minor comments

Usage of period (.)  looks strange throughout the manuscript. Most of the periods (.) should be replaced by commas (,). The manuscript contains non-English terms in table 3 as well. These make this manuscript really difficult to read.

The abstract needs to briefly describe intervention contents and actual data such as the number of participants, and between-group differences in primary outcomes at specified time frame. It is difficult to find the study's significance in the current form of abstract.

When were data collection and analyses conducted? Describe them in the trial design section.

Is GDS < 20pts correct, instead of <11pts according to the information registered?

Line 307; qualitative -> quantitative?